# A Comparison of the Ability of Some Commercially Produced Biological Control Agents to Protect Strawberry Plants against the Plant Pathogen *Phytophthora cactorum*

**Matěj Pánek [1,*], Aleš Hanáček [2], Jana Wenzlová [2], Marie Maňasová [2] and Miloslav Zouhar [2]**

[1] Team of Ecology and Diagnostics of Fungal Plant Pathogens, Crop Research Institute, Drnovská 507/73, 161 06 Prague, Czech Republic

[2] Department of Plant Protection, Faculty of Agrobiology, Food and Natural Resources, Czech University of Life Sciences in Prague, Kamýcká 129, 165 00 Prague, Czech Republic; aleshanacek@email.cz (A.H.); wenzlova@af.czu.cz (J.W.); manasova@af.czu.cz (M.M.); zouhar@af.czu.cz (M.Z.)

* Correspondence: panek@vurv.cz

**Abstract:** A comparison of the ability of commercially produced biological control agents—Contans, Gliorex, Hirundo, Polyversum, Prometheus, Clonoplus, Integral Pro and Xilon GR, completed with an isolate of *Clonostachys rosea* and of *Pseudomonas* sp.—to protect strawberry plants against *Phytophthora cactorum* was performed. The experiment was performed on strawberry cultivars Sonata, Karmen, and Wendy—cultivated in a cultivating room and greenhouse. The health of plants was affected negatively by the pathogen in all variants of biological agents used, but differences were seen in the rates of this decrease. The results revealed the ability of some tested agents to improve the growth of plants in the absence of the pathogen; the preparation Polyversum (*Pythium oligandrum*) was the most beneficial, in both the presence and absence of the pathogen. Contrarily, some agents alone decreased the health of plants; Integral Pro (*Bacillus subtillis*) and a strain of *Pseudomonas* sp. caused a deterioration in the health of the plants, even in the absence of a pathogen. The results of our analysis demonstrate the varied usefulness of all agents under unified environmental conditions; their effect seems to be dependent on the conditions and on the combination of the genotypes of all three participants in the interaction: plant–pathogen–antagonist.

**Keywords:** biological protection; black root rot; biological control agents; Pythiaceae

## 1. Introduction

*Phytophthora cactorum* (Lebert & Cohn) J. Schröt. is one of the most important pathogens of strawberry plants. Once introduced into the field, this species causes rotting in roots and plant crowns, which results in extensive economic losses. The ability of this species to spread in soil and water is immense, especially in cold and wet weather. In conditions favorable for spreading, this species creates huge numbers of actively movable zoospores [1,2] that attack new host plants; they are also able to use irrigation systems for spreading infection. In addition, their ability to persist in soil is striking; sexual and asexual reproductive structures such as oospores and chlamydospores are able to persist in soil and plant debris for many years [3–5].

Chemical compounds are routinely used to suppress infection with *P. cactorum*, although this species and other *Phytophthora* spp. have demonstrated an ability to create resistance against those compounds. The resistance of *P. cactorum* has been documented against dimethomorph [6,7], metalaxyl (mefenoxam) [8,9], cymoxanil [7], and mancozeb [10]; resistance to other compounds has been described in many related *Phytophthora* spp. [11–16]. The creation of resistance can be a quite quick process—the frequency of strains

of *P. cactorum* resistant to metalaxyl in some populations reached up to 80% as early as four years after metalaxyl began to be used in strawberry plant protection [17].

Although the use of fungicidal compounds still remains an effective method of protecting plants, in both systemic and eradicant applications, the threat of the creation of resistance against those compounds has led to efforts to find alternative means of plant protection that do not pose such a risk. Apart from the use of developed antiresistant strategies, which decrease the rate of resistance creation in the practical use of fungicides, other methods of plant protection are being tested, often based on live biological control agents (BCAs) or their metabolites. Among the organisms most used against various pathogens of the genus *Phytophthora* are fungi of the genus *Trichoderma* [18–20]; *Aureobasidium pullulans* (yeast fungus) [21]; bacterial species of the genus *Bacillus*, namely *B. amylolyquefaciens, B. megaterium, B. subtilis, B. velezensis, B. licheniformis,* and *B. cereus* [18–20,22–28]; *Pseudomonas*, namely *P. fluorescens* and *P. syringae* [19,29]; *Enterobacter cloacae*; and *Serratia ficaria* [30]. In addition, the use of arbuscular mycorrhizal fungi of genera *Glomus* and *Claroideoglomus* was successfully tested against *P. cactorum* [31]. Although many diverse tests have been performed in vitro, or in a field with individual BCAs, their routine use in the practical protection of plants has not yet been achieved.

Despite their successful use in many cases, the results of other tests show the inability of some BCAs or particular microbial strains as their active component to protect plants against fungal pathogens, or even ability to damage the health of plants, suggesting insufficient reached knowledge of the interactions of microbes and plants and the mode of action of such interaction. The fungus *Clonostachys rosea* was repeatedly described as being able to improve the protection of plants against fungal pathogens [32,33] and the oomycetes such as *Phytophthora palmivora* [32] and related *Pythium tracheiphilum* [34]. But the effect of this species on plant growth is described as both stimulating [35] and damaging [36]; some isolates were acquired from necrosis on *Pinus ponderosa* needles [37], the roots of *Pinus radiata* are protected against *Fusarium circinatum* by this fungus [38], and this species is also known as a mycoparasite, entomo- and nemato-pathogen [39,40]. A similarly ambiguous example is the use of the bacterial species *Bacillus subtillis* and other members of this genus, which has often been successfully used as a protection against plant pathogens, causing damage to their hyphae through hydrolytic enzymes [22,26,29,41]. In another case, the stimulating effect on plant growth caused by the production of indole acid was recorded [27]. However, Oh et al. [25] recorded some *Bacillus* spp. isolates' failure in plant protection against some species of the genera *Pythium* and *Phytophthora*, although the efficiency against some other tested related species was good. Some members of another bacterial genus, *Pseudomonas*, were documented as capable of inhibiting the infection of plants by releasing antibiotic compounds against oomycete pathogens such as *Pythium myriotylum* [42]; this capability was also recorded on strawberry plants against *P. cactorum* and other *Phytophthora* spp. [19,43,44]. However, other works also document the failure of *Pseudomonas fluorescens* to be protective against *P. cactorum* infection on strawberries [31] or the negative effect on plant growth of *P. aeruginosa* [45]. The mode of action of *Coniothyrium minitans* is questionable. This fungus is often mentioned in association with the protection of plants against the pathogen *Sclerotinia sclerotiorum*, whose sclerotia should be attacked by *C. minitans* [46,47]. Mycoparasitism is considered the main mode of action of this fungus [48], although some studies have mentioned antibiosis as another possibility [49,50].

Mutual comparison of the effectiveness of BCAs under unified conditions against one pathogen is quite infrequent. The objective of this study was to test the ability of commercially produced BCAs to improve the resistance of strawberry plants against *P. cactorum*. Since significant differences were documented between the results of in vitro tests on dual cultures of BCAs and the pathogen and the results of field tests [25,28], we decided to use tests on plants in controlled conditions in a cultivating room and experimental greenhouse. The experiment was supplemented by an isolate of *Pseudomonas* sp. isolated from

the colony of *P. cactorum*, of which growth was suppressed, and by an isolate of *Clonostachys rosea*; in order to relate the efficiency of BCAs to the effect of commonly used fungicides, Aliette (fosetyl-Al) was included in the tests.

## 2. Materials and Methods

### 2.1. The Growth Test of BCAs

The whole experiment was divided into two parts: the first part was performed in a growing chamber, while the second one was performed in a greenhouse. Except for the cultivating space, both parts were set up in the same way; all variants in both parts of the experiment were cultivated at the same time. Among the tested strawberry cultivars were Sonata, Wendy (in both parts of the experiment), and Karmen (only in a greenhouse). On each cultivar, the whole spectrum of BCAs was tested (Gliorex, Clonoplus, Contans, Hirundo, Prometheus, Polyversum, Integral Pro, Xilon GR, the strain of *Pseudomonas* sp. isolated from a suppressed colony of *P. cactorum*, the strain of the species *Clonostachys rosea* 156 provided from the VURV-F collection of the Crop Research Institute, the fungicidal compound Aliette, and an untreated control). Each combination of cultivar/BCA was tested on ten plants in variants with and without the pathogen. To set up the experiment, 500 mL plant pots were half filled with a cultivating substrate (the mixture of peat, compost, NPK fertilizer including microelements), the part intended to be an infection variant was immediately inoculated with a homogenate of mycelium of *P. cactorum* according to the method described in [51,52], and the pots were completed with the substrate. To achieve the highest possible infection pressure, a mix of ten *P. cactorum* isolates previously shown to be rather aggressive against strawberry plants was used; all of the included isolates we previously isolated from strawberry plants in the Czech Republic (Table 1). The species identity of isolates was determined using the metabarcoding of the ITS region of rDNA after morphological determination based on the shape of reproductive structures. The sequences of the ITS region used in identification were deposited into NCBI GenBank (Table 1). All isolates were deposited in the Collection of Agriculturally Important Fungi (VURV-F). Those isolates were cultivated in 1000 mL Erlenmeyer flasks in V8 broth for one week at 22 °C. After cultivation, the entire content of the flasks was filtered through sterile cheesecloth, and the mycelium was homogenized with an Ultra-Turrax homogenizer (15,000 rpm) in autoclaved demineralized water. An inoculum prepared in this way was used in the inoculation of the substrate in the plant pots. The total amount of inoculum was determined to be 0.2 g of dried matter for each plant pot. After 24 h, all variants were treated with specific BCAs. The individual BCAs were applied in the dilution recommended by the producers; the list of BCAs tested and the exact numbers of propagules used in the treatment of the plants are given in Table 1. The commonly employed fungicide Aliette was also included among the variants; the seedlings were immersed in a 1% suspension for 20 min just before planting. Clean water was used in control variants instead of the inoculum of the pathogen or BCAs. Frigo plants of the strawberry cultivars Sonata, Wendy, and Karmen were planted in each plant pot one week after the BCAs; the seedlings were stored at 5 °C for one week before planting. During the cultivation, the plants were arranged randomly, separately in pathogenic and nonpathogenic variants to precisely prevent the transfer of pathogen to nonpathogenic variants. The plants were then cultivated in a greenhouse or cultivating room in a 12 h/12 h photoperiod regime for eight weeks. After cultivation, all plants in each variant were removed from the pots, the substrate was carefully washed out, and the plants were dissected. Their number and the fresh weight of leaves, fruits, and blooms; the diameter and weight of the rhizome; and the length and weight of the roots were recorded.

**Table 1.** Microorganisms used in experiment.

| BCA | Active Compound | Dilution (g/mL) | No of Spores (cells)/Pot |
|---|---|---|---|
| Gliorex | *Trichoderma* sp., *Clonostachys* sp. | 2 | 1.7 mil |
| Clonoplus | *Clonostachys rosea* | 2 | 0.3 mil |
| \ | *Clonostachys rosea isolate 156* | \ | 0.3 mil |
| Contans | *Coniothyrium minitans* | 0.2 | 3.3 mil |
| Hirundo | *Bacillus* sp. | 0.05 | 0.8 mil |
| Prometheus | *Pseudomonas* | 0.05 | 0.8 mil |
| \ | *Pseudomonas* sp., *isolate* | \ | 0.8 mil |
| Polyversum | *Pythium oligandrum* | 0.15 | 2500 |
| Integral Pro | *Bacillus amyloliquefaciens (MBI 600)* | 0.05 | 18.3 mil |
| Xilon | *Trichoderma asperellum kmen T34* | 0.5 | 83,000 |

| IDs of *Phytophthora cactorum* Isolates Used | NCBI GenBank Accession Numbers | Locality of Origin |
|---|---|---|
| 17_04_12 | MW193106 | Kunratice/Central Bohemia |
| 18_10_14a | MW193116 | Břežany II/Central Bohemia |
| 17_12_20 | MW193108 | Plzeň/West Bohemia |
| 17_15_10b | OK448179 | Holešov/South Moravia |
| 17_23_19 | MW193104 | Lomec/West Bohemia |
| 19_28_2 | OK257676 | Přelovice/East Bohemia |
| 19_28_10 | OK257674 | Přelovice/East Bohemia |
| 17_30_18 | OK257655 | Svárov/North Bohemia |
| 17_34_7 | OK257657 | Lysá nad Labem/Central Bohemia |
| 17_45_1b | OK257664 | Veselá u Semil/East Bohemia |

For BCA preparations, the total dilution used in water and the resulting number of propagules per cultivating pot for treatment of each plant are given. For *Phytophthora cactorum* isolates, their IDs and the locality of origin are given. All isolates originate in strawberry plants.

*2.2. Discriminant Analysis—Statistical Analysis of Multicriterial Data*

The importance of all measured variables (the number and fresh weight of leaves, fruits, and blooms; the diameter and weight of the rhizome; and the length and weight of roots) was evaluated by discriminant analysis using Statistica 13.3 (Tibco Software Inc., Palo Alto, CA, USA). The goal of this analysis was to ascertain which of the measured variables and which combination of diverse independent variables (strawberry cultivar, cultivating space, type of BCA, presence of the pathogen) used as a grouping variable contribute the most to the differentiation between variants with and without the pathogen. The complete dataset was divided into subsets that combined the differentiation according to the two factors used—cultivating space (greenhouse/cultivating room) and strawberry cultivar (Karmen/Wendy/Sonata); each subset included both variants with and without the pathogen. In total, nine partial subsets were used in the test. One, or a combination of two or three, of the following variables was used as a grouping (independent) variable: strawberry cultivar, presence of BCAs, presence of the pathogen, and cultivating space used; a total of eight different groupings were tested. The analysis resulted in the values of the total Wilks lambda ($\lambda_W$), which expresses the ability of the model to discriminate between variants (0 = the best discriminating ability of the model, 1 = no discriminating ability of the model). For all measured variables, the *p*-values of partial $\lambda_W$, which express the importance of relevant variables for the discriminating ability of the model, were recorded. From all tested grouping models, the one having the lowest total $\lambda_W$ for all data subsets, which marks the highest distinguishing ability, was chosen. The measured variables which most contributed to differentiation using this grouping were chosen. Such variables were considered to be those whose *p*-value of partial $\lambda_W$ in all data subsets tested by this grouping model was less than or equal to 0.05.

*2.3. Principal Component Analysis—Visualization of Differences between Variants*

The differences between the compared variants based on measured multicriterial data were visualized using the results of the principal component analysis (PCA). The calculation was performed for each combination of strawberry cultivar/tested BCA/cultivating space, including all ten measured variables. The presence/absence of a pathogen was used as a grouping variable. The measured multidimensional data were recalculated to factors that well represented the distances between individuals. The results were visualized using the factors most covering the variability of data. The charts were visually evaluated to differentiate between clouds of points representing infected (P) and uninfected (C) samples. The differentiation was rated by marks between 1 (high differentiation) and 5 (no differentiation); results were recorded in the table together with the rate of variability covered by the most important calculated factors.

*2.4. A Comparison of the Most Important Variables for Differentiating between the Influence of Each BCA and of the Pathogen on Plant Growth*

The two measured variables previously chosen in the discriminant analysis as the most important in differentiation between variants—"the fresh weight of roots" and "the fresh weight of leaves"—were used in the subsequent analysis. To allow for a comparison of the total influence of each BCA and of the pathogen, the values of those two variables were averaged across data measured in a greenhouse and cultivating room for all three strawberry cultivars. The use of the weight of the roots and leaves enabled the calculation of the ratio of underground to aboveground parts of plants (i.e., a ratio resembling the roots:shoots, i.e., R:S ratio). Such a calculation neglected the weight of the rhizome and some minor parts of the plants; however, these were evaluated as less important for differentiation in the previous analysis. This ratio was calculated for each BCA for variants with and without the pathogen. Since both the pathogen and BCAs in different combinations diversely influenced the growth of the plants, the different variants combining their presence and absence were compared in order to ascertain their true influence on the plants. Therefore, the average values of the two measured variables and their ratio for each BCA of the concerned variant were expressed in percentage related to the control variant, in combinations (a–e) given in Table 2.

**Table 2.** The arrangement of five comparisons of strawberry plants of different variants of the experiment combining the presence and absence of BCAs and *P. cactorum*.

| Comparison Mode | Assessed Variant | | Control Variant | |
| :---: | :---: | :---: | :---: | :---: |
| | Presence of BCA | Presence of Pathogen | Presence of BCA | Presence of Pathogen |
| a | P | A | A | A |
| b | P | P | A | A |
| c | P | P | A | P |
| d | P | P | P | A |
| e | P | A | A | P |

In the scheme of the five comparisons (a–e), the variants of strawberry plants treated by different combinations of presence of *P. cactorum* and BCAs are given (P—present, A—absent). Each comparison was performed separately for roots, shoots, and their ratio; the comparisons were performed individually for each BCA. In the comparison, the values of the assessed variant were expressed as percentages in relation to the control variant.

## 3. Results

### 3.1. Discriminant Analysis

Discriminant analysis (DA), comparing the ability of seven groupings combining one, two, or three independent variables (strawberry cultivar, presence of BCAs, presence of the pathogen) on nine different partial datasets, resulted in the wide scale of total $\lambda_W$ (Table 3). The best distinguishing ability of the models was achieved using the groupings combining the presence of the pathogen and BCA, regardless of the particular strawberry cultivar or type of cultivating space (greenhouse/cultivating room). The distinguishing ability decreased to moderate values if the grouping exploiting the presence of BCA alone was used, while the distinguishing ability of the model based on the presence of the pathogen alone was even lower. Such results show that the type of BCA is important for the growth of strawberry plants because it increases the differences between pathogen-treated and untreated variants and between different BCA treatments. Since some of the BCAs used could have a negative impact on the strawberry plants' growth, while the impact of others would be positive, the differences between BCAs are more distinct than those among variants treated and untreated with the pathogen. Therefore, the most successful models added up the combined effect of both the pathogen and particular BCA.

Another grouping with a high differentiating ability was associated with differentiation between strawberry cultivars. Such differentiation was more distinct for the greenhouse part of the experiment, and the addition of other variables (presence of a BCA and/or pathogen) led to the model having a more precise distinguishing ability, represented by a decrease in $\lambda_W$. In the cultivating room, the distinction between cultivars was moderate, and the addition of further variables only partly refined the distinguishing ability of the model. Since all the conditions for both parts of the test were identical, except for the space used for cultivation, such results indicate the substantial influence of environment on the quality of the interaction between the plant, pathogen, and its potential antagonist.

An important result of the discriminant analysis was the identification of the two measured variables that contributed most significantly to the distinction between the tested plants treated in a different way. Such identified variables were characterized by low partial $\lambda_W$ in all partial analyses evaluated using the same grouping set-up and by the $p$-value ≤ 0.05 of the corresponding measured variable in all partial datasets tested using such grouping (Table 3). Of all the measured variables, only the fresh weight of the roots and the fresh weight of the leaves were chosen for subsequent analysis since only their $p$-values of partial $\lambda_W$ were ≤ 0.05 for all nine datasets evaluated using the most successful grouping (pathogen + BCA), although some other variables were also identified as powerful in DA.

**Table 3.** The set-up and the results of discriminant analyses.

| Partial Dataset Used (Cultivating Space/Strawberry Cultivars) | Combination of Independent Variables Used for Grouping | $\lambda_W$ | *p*-Values of Partial $\lambda_W$ of Measured Variables | | | | | | | | | |
|---|---|---|---|---|---|---|---|---|---|---|---|---|
| | | | Number of Fruits | Number of Blossoms | Weight of Fruits | Weight of Blossoms | Weight of Roots | Weight of Rhizomes | Diameter of Rhizomes | Length of Roots | Number of Leaves | Weight of Leaves |
| G/S + K + W | BCA | 0.63764 | 0.01 | 0.83 | 0.02 | 0.46 | 0.00 | 0.00 | 0.00 | 0.00 | 0.16 | 0.00 |
| G/K | BCA | 0.36297 | 0.26 | 0.71 | 0.96 | 0.84 | 0.01 | 0.00 | 0.02 | 0.00 | 0.29 | 0.02 |
| G/S | BCA | 0.30766 | 0.25 | 0.42 | 0.03 | 0.07 | 0.00 | 0.00 | 0.00 | 0.00 | 0.17 | 0.07 |
| G/W | BCA | 0.34125 | 0.02 | 0.79 | 0.00 | 0.72 | 0.02 | 0.42 | 0.00 | 0.24 | 0.20 | 0.00 |
| C/S + W | BCA | 0.63135 | 0.35 | 0.06 | 0.00 | 0.02 | 0.00 | 0.13 | 0.16 | 0.32 | 0.21 | 0.31 |
| C/S | BCA | 0.40998 | 0.30 | 0.18 | 0.00 | 0.66 | 0.55 | 0.32 | 0.08 | 0.01 | 0.03 | 0.17 |
| C/W | BCA | 0.45809 | 0.76 | 0.20 | 0.55 | 0.20 | 0.00 | 0.07 | 0.16 | 0.85 | 0.03 | 0.17 |
| G/S + K + W | Cultivar | 0.27766 | 0.00 | 0.00 | 0.00 | 0.00 | 0.02 | 0.00 | 0.00 | 0.00 | 0.00 | 0.00 |
| C/S + W | Cultivar | 0.60780 | 0.00 | 0.13 | 0.01 | 0.26 | 0.08 | 0.01 | 0.00 | 0.00 | 0.40 | 0.00 |
| G/S + K + W | Cultivar + BCA | 0.09009 | 0.00 | 0.01 | 0.00 | 0.01 | 0.00 | 0.00 | 0.00 | 0.00 | 0.00 | 0.00 |
| C/S + W | Cultivar + BCA | 0.60780 | 0.00 | 0.13 | 0.01 | 0.26 | 0.08 | 0.01 | 0.00 | 0.00 | 0.40 | 0.00 |
| G/S + K + W | Cultivar + Pathogen | 0.20683 | 0.00 | 0.00 | 0.00 | 0.00 | 0.01 | 0.00 | 0.00 | 0.00 | 0.00 | 0.00 |
| C/2 | Cultivar + Pathogen | 0.35366 | 0.00 | 0.00 | 0.00 | 0.00 | 0.00 | 0.01 | 0.00 | 0.02 | 0.48 | 0.00 |
| C/2 | Cultivar + Pathogen + BCA | 0.06205 | 0.00 | 0.00 | 0.00 | 0.00 | 0.00 | 0.00 | 0.00 | 0.01 | 0.00 | 0.00 |
| G/S + K + W | Pathogen | 0.88373 | 0.00 | 0.02 | 0.00 | 0.00 | 0.02 | 0.00 | 0.61 | 0.40 | 0.06 | 0.00 |
| G/K | Pathogen | 0.84736 | 0.65 | 0.22 | 0.10 | 0.96 | 0.21 | 0.01 | 0.60 | 0.07 | 0.01 | 0.00 |
| G/S | Pathogen | 0.62026 | 0.00 | 0.00 | 0.00 | 0.25 | 0.28 | 0.62 | 0.92 | 0.32 | 0.83 | 0.00 |
| G/W | Pathogen | 0.77976 | 1.00 | 0.80 | 0.56 | 0.40 | 0.24 | 0.00 | 0.13 | 0.42 | 0.20 | 0.73 |
| C/S + W | Pathogen | 0.66781 | 0.02 | 0.00 | 0.00 | 0.00 | 0.00 | 0.09 | 0.30 | 0.12 | 1.00 | 0.00 |
| C/S | Pathogen | 0.64577 | 0.10 | 0.00 | 0.00 | 0.00 | 0.60 | 0.72 | 0.14 | 0.42 | 0.31 | 0.10 |
| C/W | Pathogen | 0.60210 | 0.25 | 0.38 | 0.01 | 0.02 | 0.00 | 0.26 | 0.44 | 0.26 | 0.98 | 0.00 |
| G/S + K + W | Pathogen + BCA | 0.39665 | 0.00 | 0.09 | 0.00 | 0.01 | 0.00 | 0.00 | 0.00 | 0.00 | 0.05 | 0.00 |
| G/K | Pathogen + BCA | 0.15278 | 0.41 | 0.45 | 0.38 | 0.86 | 0.01 | 0.00 | 0.05 | 0.00 | 0.00 | 0.00 |
| G/S | Pathogen + BCA | 0.04398 | 0.00 | 0.00 | 0.00 | 0.01 | 0.00 | 0.00 | 0.00 | 0.00 | 0.28 | 0.00 |
| G/W | Pathogen + BCA | 0.09164 | 0.04 | 0.92 | 0.00 | 0.78 | 0.02 | 0.00 | 0.00 | 0.00 | 0.26 | 0.02 |
| G + C/S | Pathogen + BCA | 0.23271 | 0.74 | 0.00 | 0.00 | 0.12 | 0.00 | 0.01 | 0.00 | 0.00 | 0.01 | 0.00 |
| C/S + W | Pathogen + BCA | 0.27867 | 0.28 | 0.00 | 0.00 | 0.00 | 0.00 | 0.03 | 0.54 | 0.05 | 0.08 | 0.00 |
| C/S | Pathogen + BCA | 0.10628 | 0.40 | 0.00 | 0.00 | 0.05 | 0.04 | 0.09 | 0.07 | 0.02 | 0.06 | 0.05 |
| C/W | Pathogen + BCA | 0.12617 | 0.92 | 0.29 | 0.57 | 0.34 | 0.00 | 0.21 | 0.35 | 0.39 | 0.01 | 0.00 |
| G + C/W | Pathogen + BCA | 0.30120 | 0.25 | 0.23 | 0.15 | 0.23 | 0.00 | 0.01 | 0.00 | 0.00 | 0.00 | 0.00 |
| G + C/S | Cultivating space | 0.42850 | 0.02 | 0.93 | 0.50 | 0.00 | 0.00 | 0.00 | 0.00 | 0.02 | 0.00 | 0.10 |
| G + C/W | Cultivating space | 0.31110 | 0.51 | 0.00 | 0.02 | 0.91 | 0.00 | 0.00 | 0.00 | 0.01 | 0.00 | 0.00 |

The set-up of partial datasets in combination with particular grouping used in discriminant analyses is given. The used partial datasets combine the data from particular cultivating space and from each of tested strawberry cultivars (cultivating space: G—greenhouse, C—cultivating room; strawberry cultivars: K—Karmen, S—Sonata, W—Wendy). Combinations of independent variables—the presence/absence of pathogen, the type of BCA, and the tested strawberry cultivar—were used as a grouping variable. The Wilks $\lambda$ ($\lambda W$) values are given for each tested combination of partial dataset and grouping. For each measured variable, the *p*-value related to its partial $\lambda W$ is given.

### 3.2. Principal Component Analysis

Using this method, a total of ten factors (axes) were identified, which were used to project the measured data, thus covering 100% of the variability. Taking all tested cases into consideration, the first factor covered 29.13–73.77% and the second one covered 11.91–28.05% of the variability. In total the first two factors covered 50.10–85.68% of the variability found between strawberry plants in each particular BCA variant including both plants treated and untreated by pathogen. The rest of the variability was divided between eight other factors (Table 4). In the resulting charts, therefore, these two main factors were used to project the relations among variants with (P) and without (C) the pathogen (Figure 1, Supplementary Figure S1). The case of clear differentiation of the clouds of P and C points in charts is interpretable as the result of different growth and health of infected plants in particular BCA variant as a consequence of pathogen infection. Therefore, such tested BCA cannot be considered as an agent providing plant protection. Contrarily, the overlap of these two clouds means low or no differentiation between plants caused either by the protective effect of the BCAs on infected plants (Figure 1a) or by the negative influence of BCA alone, which decreases the plant growth in a similar range as a pathogen (Figure 1b). The gradual tendency to differentiate is visible on the chart examples in Figure 1a–d, documenting the different abilities of tested BCAs to protect strawberry plants against *P. cactorum*.

In all comparisons of samples in combinations of the strawberry cultivar/tested BCA/cultivating space, the strong differentiation of plants infected by the pathogen from those uninfected was obvious, particularly for the variants Aliette and *Clonostachys rosea* isolate 156 and in some combinations with Gliorex, Prometheus, Xilon GR, and Contans. On the contrary, little or no differentiation was found in variants treated with Polyversum, Integral Pro, Clonoplus, Contans, and an isolate of *Pseudomonas* sp. Since PCA does not provide information quantifying growth, a mutual comparison of the results of each particular BCA is not possible.

The results of the PCA also enabled a rough comparison of the effect of BCAs on infection for the respective strawberry cultivars—unlike the other two variants, no differentiation among infected and uninfected variants was recorded in all BCA variants in the cultivar Karmen (Table 4). However, barring the effect of BCAs, this could alternatively be attributed to the high resistance of this cultivar to infection with *P. cactorum* [53].

**Table 4.** The results of evaluation of principal component analysis (PCA) charts.

| Cultivating Space Used | Strawberry Cultivar | BCA Tested | Covered Variability (%) | | | Estimated Differentiation Read from Plots |
|---|---|---|---|---|---|---|
| | | | Factor 1 | Factor 2 | Sum of the First Two Factors | |
| C | S | Aliette | 62.58 | 15.94 | 78.52 | 2 |
| C | W | Aliette | 47.79 | 16.48 | 64.27 | 1 |
| C | S | Integral Pro | 47.03 | 21.6 | 68.63 | 5 |
| C | W | Integral Pro | 44.58 | 17.96 | 62.54 | 5 |
| G | S | Integral Pro | 41.96 | 19.57 | 61.53 | 5 |
| G | W | Integral Pro | 42.14 | 24.24 | 66.38 | 3 |
| C | S | Clonoplus | 55.81 | 14.78 | 70.59 | 3 |
| C | W | Clonoplus | 47.5 | 17.13 | 64.63 | 5 |
| G | S | Clonoplus | 51.29 | 22.2 | 73.49 | 4 |
| G | K | Clonoplus | 34.92 | 23.74 | 58.66 | 5 |
| G | W | Clonoplus | 38.59 | 18.87 | 57.46 | 5 |
| C | S | *Clonostachys rosea* isolate 156 | 54.01 | 17.54 | 71.55 | 2 |
| C | W | *Clonostachys rosea* isolate 156 | 58.62 | 15.58 | 74.2 | 1 |
| C | S | Contans | 55.34 | 21.27 | 76.61 | 5 |
| C | W | Contans | 46.03 | 21.31 | 67.34 | 5 |
| G | S | Contans | 54.85 | 21.65 | 76.5 | 5 |
| G | K | Contans | 35.86 | 26.23 | 62.09 | 5 |
| G | W | Contans | 42.38 | 20.88 | 63.26 | 2 |
| C | S | Control | 62.72 | 17.08 | 79.8 | 5 |
| C | W | Control | 53.3 | 15.85 | 69.15 | 3 |

| | | | | | | |
|---|---|---|---|---|---|---|
| G | S | Control | 43.07 | 15.88 | 58.95 | 4 |
| G | K | Control | 35.16 | 21.76 | 56.92 | 5 |
| G | W | Control | 49.5 | 15.18 | 64.68 | 5 |
| C | S | Gliorex | 61.48 | 14.98 | 76.46 | 2 |
| C | W | Gliorex | 33.93 | 22.22 | 56.15 | 4 |
| G | S | Gliorex | 41.4 | 28.05 | 69.45 | 5 |
| G | W | Gliorex | 46.92 | 22.08 | 69 | 2 |
| C | S | Hirundo | 59.06 | 24.25 | 83.31 | 4 |
| C | W | Hirundo | 50.95 | 15.03 | 65.98 | 4 |
| G | S | Hirundo | 42.7 | 20.15 | 62.85 | 5 |
| G | K | Hirundo | 42.04 | 22.91 | 64.95 | 5 |
| G | W | Hirundo | 57.03 | 14.6 | 71.63 | 1 |
| C | S | Polyversum | 47.85 | 21.29 | 69.14 | 4 |
| C | W | Polyversum | 31.04 | 19.06 | 50.1 | 4 |
| G | S | Polyversum | 51.33 | 27.2 | 78.53 | 5 |
| G | K | Polyversum | 39.51 | 23.03 | 62.54 | 5 |
| G | W | Polyversum | 31.17 | 23.82 | 54.99 | 5 |
| C | S | Prometheus | 48.64 | 17.98 | 66.62 | 2 |
| C | W | Prometheus | 43.01 | 16.08 | 59.09 | 4 |
| G | S | Prometheus | 53.74 | 16.83 | 70.57 | 4 |
| G | K | Prometheus | 51.74 | 19.94 | 71.68 | 5 |
| G | W | Prometheus | 40.02 | 21.31 | 61.33 | 5 |
| C | S | *Pseudomonas* sp. | 73.77 | 11.91 | 85.68 | 5 |
| C | W | *Pseudomonas* sp. | 50.36 | 20.62 | 70.98 | 4 |
| G | S | *Pseudomonas* sp. | 37.38 | 22.99 | 60.37 | 3 |
| G | K | *Pseudomonas* sp. | 33.13 | 27.82 | 60.95 | 5 |
| G | W | *Pseudomonas* sp. | 50.93 | 20.83 | 71.76 | 5 |
| C | S | Xilon | 52.66 | 20.19 | 72.85 | 4 |
| C | W | Xilon | 49.32 | 20.4 | 69.72 | 2 |
| G | S | Xilon | 41.29 | 23.68 | 64.97 | 3 |
| G | K | Xilon | 29.13 | 20.97 | 50.1 | 5 |
| G | W | Xilon | 38.42 | 24.19 | 62.61 | 5 |

For each combination of cultivating space/strawberry cultivar/BCA tested, the estimated rate of differentiation read from PCA charts of plants infected by *P. cactorum* from those without pathogen is given; the scale is 1—highest, 5—lowest differentiation. Variability covered by the first two factors computed by PCA and their sum are shown. Cultivating space: C—cultivating room, G—greenhouse. Strawberry cultivars: S—Sonata, K—Karmen, W—Wendy.

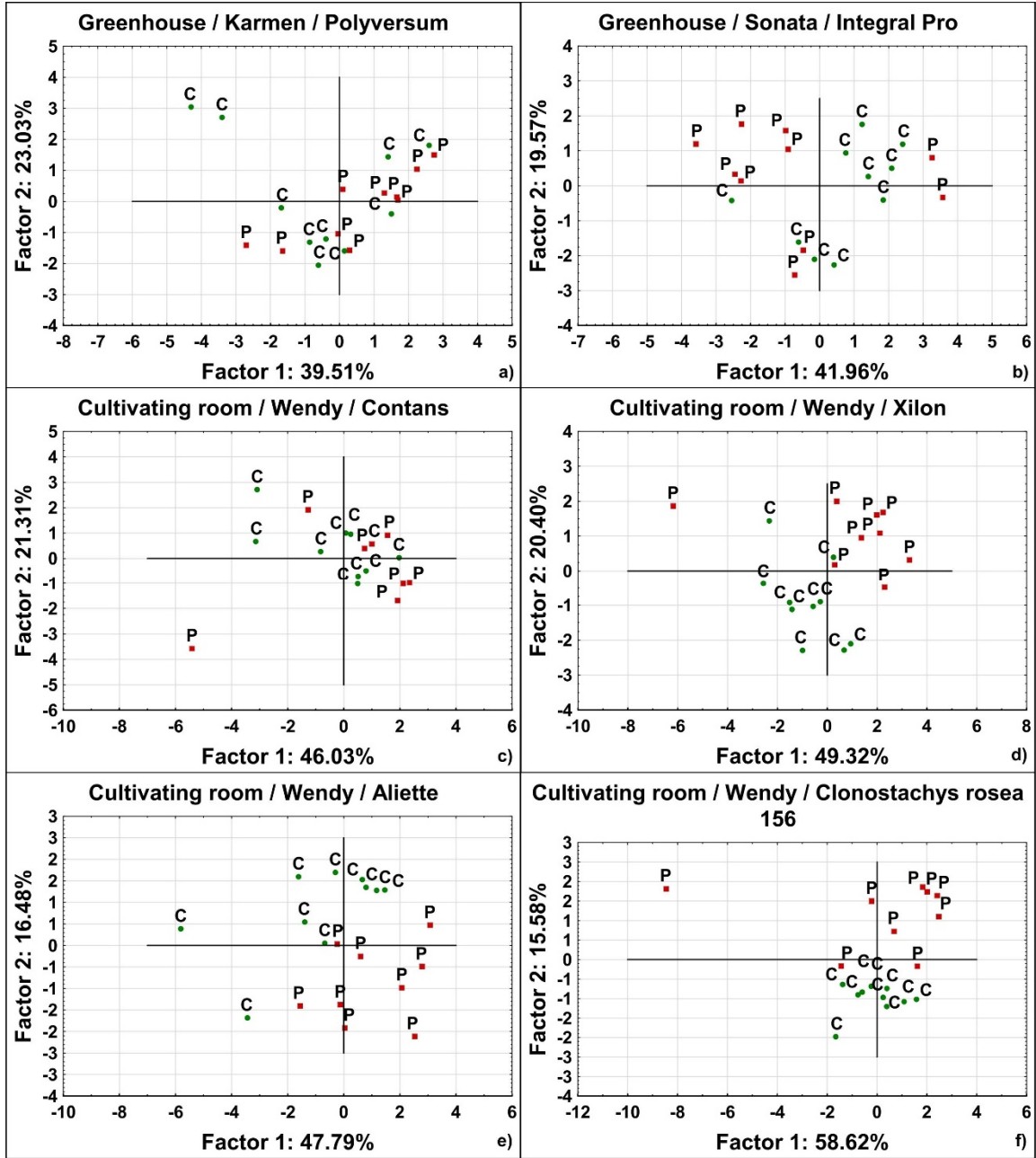

**Figure 1.** The examples of results of the principal component analysis comparing the plants infected by *Phytophthora cactorum* (P) with uninfected control plants (C), where all the plants were treated by tested BCAs. The overlap of the clouds of P and C points in the charts (**a**) is interpretable as low or no differentiation between plants caused either by the protective effect of BCAs on infected plants or by the negative influence of BCA alone, while their clear differentiation (**f**) could mean the result of different growth and different health of infected plants in consequence of pathogen infection. The subfigures (**a**–**f**) show the continuing tendency between those utmost cases.

### 3.3. A Comparison of the Weight of Roots, the Weight of Leaves, and Their Ratio

Five comparisons were performed based on two measured variables, namely the weight of roots and the weight of leaves, and on their calculated ratio R:S. All three variables were compared between different variants of the test in the set-up given in Table 2 The values of each of these variables in the evaluated variant were expressed as percentages in relation to the appropriate control variant (Tables 2 and 5). The influence of both BCA and pathogen on the weight of roots and leaves was evaluated as negative in case of their decrease in comparison to the control variant, while the decrease in the R:S ratio in this comparison was evaluated as a result of the positive influence of BCA; contrarily, the

increase was attributed to the influence of the pathogen or the negative influence of the BCA alone.

### 3.3.1. Influence of the Tested BCAs Alone on the Growth of Plants

The plants treated with BCAs were compared to those totally untreated (Table 5a). The growth of plants treated with the majority of the tested BCAs was the same or higher in comparison to untreated plants, which is obvious considering the weight of the roots and leaves, as well as their ratio. The exception was Integral Pro, which decreased the weight of the roots and leaves and increased their ratio. The growth of only roots decreased in the variants treated with an isolate of *Clonostachys rosea* 156 and with Contans; a worsening of the R:S ratio was recorded in the variant treated with Integral Pro and an isolate of *Pseudomonas* sp. The growth of leaves and the R:S ratio in the variant treated with Aliette were improved. Such results document the different influences of BCAs alone on the growth of the host plants, which could be both negative and positive.

### 3.3.2. Joint Influence of BCAs and the Pathogen on the Plants

Plants with BCAs and the pathogen were compared to the totally untreated variant (Table 5b). The growth of the variants treated with both the pathogen and BCAs mostly decreased in comparison to the untreated variant, which was more visible on leaves and on the R:S ratio than on roots. This ratio was negatively influenced for all variants; only in Polyversum did the margin not exceed 100%. The weight of the roots for several variants was higher than in the control; lower values were recorded for leaves. The weight of roots and the weight of leaves, as well as their ratio, were the same as or better than those in the control only for the variant treated with Polyversum. This comparison demonstrates that the growth of plants in all variants of BCA treatment was affected by infection with *P. cactorum*, although the total rate of growth inhibition was quite different. In the case of reduced growth caused by some BCAs in this comparison, such potential negative influence of BCAs is indistinguishable from that of the pathogen. Therefore, in such a case, the growth of the plants should be considered as some kind of summation of the effects of both these microorganisms. A comparison to the untreated control and to the variant treated with Aliette demonstrated the protective potential of some of the BCAs used.

### 3.3.3. Influence of BCAs on Plants with the Pathogen

The plants treated with a pathogen and BCAs were compared to those treated only with a pathogen (Table 5c). All three compared variables (weight of roots, weight of leaves, and their ratio) were not worse than the control without BCAs only for the variants treated with Polyversum; a quite small decrease was seen in the variants treated with Prometheus and *Clonostachys rosea* isolate 156. In several variants, the weights of both the leaves and roots were higher than in the control, although the R:S ratio increased significantly. A negative influence on all three variables was recorded only in the variants treated with Contans and Integral Pro. This result demonstrates that the growth of strawberry plants is influenced by *P. cactorum* differently according to the BCA used; however, the influence of the pathogen is mostly reduced by the majority of the BCAs, even in comparison to the variant treated with Aliette.

### 3.3.4. Influence of the Pathogen on Plants with BCAs

The plants with BCAs and the pathogen were compared to the control treated only with BCAs (Table 5d). The weight of leaves decreased in all variants, while the mass of roots increased in some variants with both BCA and the pathogen, in comparison to the control; the R:S ratio increased in all variants, signaling the worsening health of the plants. Substantial differences were found, however, in the degree of worsening ascertained in each variant; the lowest R:S was found in the variant treated with Polyversum (115%), but even in this variant, the mass of leaves and roots was affected by infection. The highest

R:S was found in the variants treated with pathogen together with *Clonostachys rosea* isolate 156 (333%) or Aliette (582%). This result documents that even in the presence of each of the tested BCAs, the presence of the pathogen leads to the worsening health of the plants, although the plants are sometimes able to respond by increasing the growth of roots. The disproportional changes in roots and shoots are associated with an increasing R:S ratio, which unambiguously documents the deterioration of plant health.

3.3.5. Comparison of the Influence of Each BCA with That of the Pathogen

The majority of the average weights of roots and leaves, as well as the R:S ratio, were better in the variants treated with BCAs compared to those treated with a pathogen (Table 5e). The differences were more distinct for leaves, for which the growth in BCA variants reached as much as hundreds of percent of the growth of the pathogen-treated variant. In this comparison, three cases of decreased root growth were also recorded (variants with *Clonostachys rosea* isolate 156, Contans, and Gliorex), although this decrease was not large. A worsening of the R:S ratio comparable to that caused by the pathogen was recorded in variants treated with Integral Pro and with an isolate of *Pseudomonas* sp., but the total mass of leaves and roots was significantly higher in both those treatments. This result documents at least the partial negative influence of some BCAs on the growth of plants, although it is rather insignificant in the majority of comparisons to the pathogen (Table 5e).

**Table 5.** The results of five comparisons of strawberry plants in different variants of combinations of the presence and absence of BCAs and *P. cactorum*.

| Comparison Mode | | (a) | | | (b) | | | (c) | | | (d) | | | (e) | | |
|---|---|---|---|---|---|---|---|---|---|---|---|---|---|---|---|---|
| Control Variant | | BCA-Free | | | BCA-Free | | | BCA-Free | | | BCA | | | BCA Free | | |
| | | Pathogen-Free | | | Pathogen-Free | | | Pathogen | | | Pathogen-Free | | | Pathogen | | |
| Assessed Variant | | BCA | | | BCA | | | BCA | | | BCA | | | BCA | | |
| | | Pathogen-Free | | | Pathogen | | | Pathogen | | | Pathogen | | | Pathogen Free | | |
| Tested BCA | Active Component | Roots | Leaves | R:S | Roots | Leaves | R:S | Roots | Leaves | R:S | Roots | Leaves | R:S | Roots | Leaves | R:S |
| Aliette | Fosetyl-Al | 99.4 | 197.4 | 51.1 | 95.1 | 43.3 | 291.0 | 99.2 | 99.2 | 258.0 | 104.0 | 29.1 | 582.9 | 98.4 | 295.5 | 36.6 |
| Integral Pro | *Bacillus amyloliquefaciens* | 94.9 | 72.4 | 141.4 | 106.2 | 40.7 | 197.1 | 98.1 | 77.1 | 166.5 | 112.6 | 56.4 | 143.1 | 111.1 | 138.9 | 103.5 |
| Clonoplus | 4 strains of *Clonostachys rosea* | 117.3 | 184.5 | 98.9 | 88.9 | 82.0 | 160.1 | 116.0 | 112.1 | 132.6 | 89.6 | 43.2 | 181.5 | 93.6 | 261.3 | 89.6 |
| *Clonostachys rosea* | *Clonostachys rosea* isolate 156 | 93.0 | 207.2 | 47.9 | 116.3 | 77.7 | 162.1 | 93.0 | 118.3 | 106.5 | 133.3 | 39.5 | 333.6 | 119.9 | 294.7 | 32.8 |
| Contans | *Coniothyrium minitans* | 88.3 | 97.1 | 97.9 | 85.1 | 50.5 | 162.8 | 87.9 | 88.2 | 141.1 | 104.0 | 52.2 | 169.4 | 87.4 | 167.6 | 89.3 |
| Control | \ | 100.0 | 100.0 | 100.0 | 97.7 | 75.5 | 127.8 | 104.4 | 100.0 | 100.0 | 97.7 | 75.5 | 127.8 | 100.0 | 183.2 | 88.0 |
| Gliorex | *Clonostachys rosea*, *Trichoderma asperellum* | 98.9 | 207.3 | 77.9 | 92.3 | 53.9 | 160.3 | 99.3 | 118.6 | 147.8 | 97.6 | 34.8 | 262.1 | 94.8 | 304.1 | 69.8 |
| Hirundo | *Bacillus amyloliquefaciens* | 106.8 | 186.2 | 98.4 | 96.5 | 72.1 | 210.1 | 107.1 | 94.6 | 177.0 | 93.0 | 35.9 | 221.8 | 98.1 | 274.9 | 89.3 |
| Polyversum | *Pythium oligandrum* | 138.0 | 201.2 | 88.5 | 107.4 | 116.0 | 98.3 | 144.5 | 201.7 | 85.4 | 77.8 | 60.6 | 115.3 | 112.3 | 316.3 | 76.4 |
| Prometheus | *Pseudomonas veronii* | 125.3 | 178.2 | 85.8 | 109.0 | 54.2 | 145.5 | 131.1 | 108.0 | 129.3 | 87.4 | 33.8 | 198.4 | 111.1 | 283.8 | 73.7 |
| *Pseudomonas* sp. | *Pseudomonas* sp. isolate | 114.1 | 143.3 | 115.7 | 108.4 | 100.8 | 306.9 | 112.6 | 147.0 | 268.7 | 112.3 | 67.6 | 231.0 | 113.3 | 234.6 | 106.4 |
| Xilon | *Trichoderma asperellum* | 139.8 | 205.1 | 102.1 | 103.8 | 102.3 | 195.4 | 143.3 | 132.2 | 173.7 | 73.8 | 45.6 | 197.6 | 105.7 | 311.1 | 90.2 |

Results of five comparisons (**a**–**e**) of averages calculated across all three cultivars Wendy, Sonata, and Karmen. The averages of the weight of roots, the weight of leaves, and their ratio (R:S) are displayed. The averages of measurements for different combinations of the presence and absence of each BCA and *P. cactorum* in assessed variants were compared to various combinations of their presence and absence in control variants. The values of assessed variants are expressed as percentages in relation to the particular control variant.

## 4. Discussion

The use of biological control agents is an increasingly mentioned possibility in plant protection against diverse pathogenic microorganisms. The use of such a method of plant protection has many potential advantages in comparison with commonly used fungicides, such as the low probability of resistance creation in the plant pathogen or the chance to avoid introducing chemical compounds into the environment and into human food. Although some BCAs have become part of commercially produced preparations and are successfully used in some cases, their influence on the plant–pathogen system is unclear.

Our results document the important influence of the tested BCAs on the growth of strawberry plants and on the development of infection with *P. cactorum*. This is documented by the results of discriminant analysis, represented by a rather low $\lambda w$ (Table 3) in the case of the use of a grouping based on a combination of treatments with BCAs and the pathogen. Such a high distinguishing ability of the test, taking all measured variables into account, demonstrates quite a diverse capability of the used BCAs to decrease the damage to plants caused by *P. cactorum*. Although marked differentiation was unquestionable, discriminant analysis does not enable the unambiguous identification of the most suitable BCAs for plant protection. The level of differentiation of infected and uninfected plants is also visible on the charts resulting from PCA analysis (Figure 1, Supplementary Figure S1, Table 4), although such differentiation unambiguously identified only the BCAs that are not able to protect plants against the pathogen and simultaneously do not have a negative influence on the plants. The analysis is not able to distinguish cases of low growth of the uninfected plants potentially caused by some BCAs from the low growth of those infected by the pathogen or cases of the high growth of plants protected by BCAs against infection with the pathogen from the high growth of plants without the pathogen. In both cases, the charts do not display any differentiation (Figure 1, Supplementary Figure S1). Even a decision based on a direct comparison of the measured variables is not reliable, since in addition to the pathogen, some of the tested BCAs also have a negative influence on plants or on their parts, even though the majority of them do not negatively affect growth in the absence of the pathogen. This is visible in the results of the discriminant analysis, where the distinguishing ability among variants increases when the joint presence of BCAs and the pathogen was used as a grouping variable, in comparison to using groupings only by the presence of the pathogen or BCAs. The influences of each BCA and the pathogen are thus summed together, and the negative and positive effect of different BCAs increases the differences among variants caused by the pathogen alone. In addition, the reaction of different parts of the plants to the presence of each of the tested BCAs and of the pathogen differs.

The results of our discriminant analysis demonstrated that the influence of both the pathogen and its potential antagonist is the most distinctively obvious regarding the weight of the leaves and the roots. With some exceptions, the influence of the majority of the BCAs alone on the growth of roots and leaves is mostly not negative (Table 5a), while their positive influence in the presence of the pathogen is less distinct (Table 5c). Almost none of the tested BCAs were able to improve the weight of the roots and leaves of plants infected by the pathogen to the level of healthy plants untreated with BCAs (Table 5b); only Polyversum, Xilon GR, and *Pseudomonas* isolate were associated with a nondecreased weight. The next comparison (Table 5d), however, demonstrates that infection with the pathogen decreases the weight of leaves in all variants and the weight of roots in the majority of variants of BCA treatment. The decrease in both those variables is unequal. The last comparison (Table 5e) brings somewhat surprising information, namely that some of the tested BCAs decrease the growth of roots in a measure comparable to that caused by the pathogen. Since the growth of leaves and roots obviously reflects the presence of both the pathogen and BCAs differently, the decision on which BCA is suitable for plant protection is also ambiguous on this basis.

A useful indicator of the health state of plants that provides the more reliable information than the weight of roots or the weight of leaves alone is their ratio. This variable

reflects the allocation of biomass into each organ in relation to its effectiveness [54], and thus its change also indicates the functional deterioration of the plant. Plants under drought stress allocate relatively more biomass to the root systems to improve water uptake [54], while an insufficiently large or effective root system is not able to supply a plant's aboveground parts that are too large, and these parts then decrease in relative size. This ratio is thus useful as a health index [55,56]. As a soil pathogen, *P. cactorum* primarily attacks the roots of plants, or the conductive tissues in the plant crown, which restricts the uptake of nutrients and particularly of water by the plants. The resulting influence on the plant is therefore similar to that in the case of drought stress, which significantly increases the R:S ratio [57,58]. The majority of our tested BCAs alone were associated with a decrease in the R:S ratio (Table 5a). The decreasing of this ratio also was apparent in the comparison of plants treated by BCAs with those treated with the pathogen (Table 5e); in almost all variants, this ratio was lower in BCA-treated variants than in those infected with the pathogen. The majority of the tested BCAs thus could be considered as improving plant health in the absence of the pathogen, with the exception of Integral Pro and the isolate of *Pseudomonas* sp. The R:S ratio decreased, or at least did not greatly increase, in plants with the pathogen when simultaneously treated with Polyversum or *C. rosea* isolate 156 (Table 5c) in comparison to infected plants without BCAs. In the case of treatment of plants with Polyversum together with the pathogen, the R:S ratio did not increase above the level of totally untreated plants (Table 5b).

Our analysis thus revealed the negative influence of the pathogen on the R:S ratio in all treatment variants (Table 5d), even in the presence of BCAs, in comparison to variants treated with the same BCAs but without the pathogen. However, the magnitude of such R:S worsening ranged between a small effect of the pathogen (Polyversum) and significant worsening of plant health (Aliette, *Pseudomonas* sp.); among the other rather less affected variants were those treated with Integral Pro (Table 5d). Although treatment with Integral Pro and Polyversum evinced similarly the lowest decrease in plant health caused by the pathogen in comparison to the others, the meaning of such a small decrease is different. In comparison to untreated plants, the plants treated merely by Integral Pro evinced decreased growth in a similar range to that caused by pathogen alone. As a control variant, therefore, a comparison of plants with Integral Pro and the pathogen to those with only Integral Pro (Table 5d) used the plants whose health was negatively affected by this BCA alone (Table 5a). The small decrease in health is therefore illusory. The plants of this variant infected by the pathogen also showed a worsening of health in comparison to infected control plants without any BCAs (Table 5b).

The variant with Polyversum represented the inverse situation. The plants treated with this BCA also showed a worsening of health caused by the pathogen in comparison to the same variant without the pathogen (Table 5d), and this worsening was of the least extent of all treatments. However, in this comparison, the infected plants were compared to healthy plants whose growth was strongly stimulated by the Polyversum treatment, which is obvious in Table 5a, and therefore the decrease associated with the pathogen seems to be large. In fact, in the presence of Polyversum, the plants affected by the pathogen are in an even better state of health than the plants without both BCAs and the pathogen (Table 5b,c). The mechanism of action of *Pythium oligandrum* as an active compound of Polyversum is not yet entirely clear and probably combines multiple action modes. The work of You et al. [59] mentioned competition for space and nutrients as a probable mechanism; other possibilities include antibiosis and parasitism [60,61] and induced resistance [62]. Our results document the strong positive influence of this BCA on the growth of the plant in the absence of the pathogen. This is visible for both roots and leaves, as well as the R:S ratio, which is rather low and does not increase excessively in the presence of the pathogen, although even with this treatment the pathogen caused some decrease in both roots and leaves. The observed growth-stimulating effect we found is explainable as a result of auxin-like compounds produced by *P. oligandrum*, mentioned by Le Floch et al. [63]. As precursors of auxin, those compounds are able to contribute to the optimization

of plant growth, which is in agreement with the balanced R:S ratio we found. The probable stimulating effect on plant growth as a mechanism of action does not rule out the current involvement of another mechanism of action in improving plant growth and resistance. However, the very possibility of a simple stimulatory effect on plant growth that outweighs the negative influence of the pathogen provides an explanation for the better health of the infected plants. The magnitude of such an improving effect of *Pythium oligandrum* on strawberry plants significantly exceeds the effect of all other tested BCAs.

In our study, we also tested *Clonostachys rosea* in three preparation forms—the isolate No. 156 from the collection VURV-F, the active component of the preparation Clonoplus, and together with *Trichoderma* sp. as a component of the Gliorex preparation. Our results showed that all three tested BCAs containing the inoculum of this fungus had a rather positive effect on the growth of strawberry plants. However, none of the three preparations we used in our test demonstrated an ability to improve the growth of strawberry plants in the presence of *P. cactorum*, although their effects were diverse.

Two of the preparations we tested (Xilon GR and Gliorex) contain the inoculum of members of the genus *Trichoderma*. Although many members of this genus are mentioned as being effective against *Phytophthora* pathogens [64–67], our results did not confirm the usefulness of Gliorex or Xilon GR for the protection of strawberry plants against *P. cactorum*, while the growth of uninfected plants was influenced quite positively. In different comparisons of the influence of those two BCAs (Table 5b,c), Xilon GR seemed to be more effective in increasing plant biomass than Gliorex, although the R:S ratio always negatively deviated in the presence of the pathogen in both of those two BCA variants.

In contrast to the otherwise rather beneficial influence of BCAs on the growth of plants, discussed above, was the already mentioned influence of Integral Pro, which negatively impacted plants in both the presence and absence of the pathogen despite frequently mentioned successful use of the active component of this BCA, i.e., *Bacillus subtillis*, and other members of this genus in plant protection [22,26,29,41].

A negative effect on plants was also found in variants treated with an isolate of *Pseudomonas* sp., whose influence on the weight of leaves or roots alone was slightly negative; however, the R:S ratio of these variants increased even in the absence of the pathogen and strongly increased in its presence. Less significant was the effect of the preparation Prometheus, whose active component is *Pseudomonas veronii*. Even this BCA was not able to protect plants against the pathogen, but the influence on the growth of plants alone was positive.

Another BCA with a quite negative influence on plant growth was the preparation Contans, whose active compound is the fungus *Coniothirium minitans*. Our results did not confirm an important protective effect against *P. cactorum*, while the growth of plants treated with Contans alone was rather worse in comparison to those untreated, which indeed suggests the production of some plant growth modulating metabolites by this BCA as already described Tomprefa et al. [48].

The majority of the BCAs we tested did not impact plant growth negatively, and at least 5 of the 10 tested can be considered beneficial for plant growth. This is especially obvious for the example of Polyversum. A stimulating effect [22,27,68] and inhibition of growth [27,29,69,70] and plant cell death [71] have been documented in diverse BCAs. As mentioned by Blom et al. [68], the resultant quality of interaction between microorganism and plant may be dependent on the concentration of microbial dosage; their release of some volatile components and their concentration alone also strongly depend on the nutritional richness of the cultivating media [25,31,68,71,72]. Similarly, the influence of the type of cultivating media on the quality of interaction between *Trichoderma harzianum* and *Phytophthora capsici* was uncovered [67]. Although proven to be effective against *P. cactorum* in our tests, the effect of *Pyhtium oligandrum* also seems to be dependent on environmental conditions [73]. The quality of interaction of the whole complex of plant, pathogen, and antagonist seems to be dependent on the environmental conditions, which is also congruent with our results. Some differences were revealed between the results of our

tests performed in the greenhouse and in the cultivating room, although except for the type of cultivating space all other conditions were identical. This is demonstrated by medium values of Wilks $\lambda$ in discriminant analysis (0.3111 for cultivar Wendy and 0.4285 for Sonata, Table 3) using the cultivating room as the only grouping variable. Our results thus indicate the quite different usefulness of diverse BCAs in identical conditions. Although we identified only Polyversum as unambiguously able to protect strawberry plants against *P. cactorum*, the abiotic conditions of the environment and even the presence of other microorganisms are probably able to modify the efficiency of each BCA. Our results thus should be considered only in relation to the narrow conditions we used for plant cultivation, and changes in such conditions probably would impact the efficiency of particular BCAs.

The efficiency of BCAs as protective against pathogens has already been shown to be associated with the particular genotype of the microorganism and host plants used. This was demonstrated in the case of *Pseudomonas fluorescens* [42], *Clonostachys rosea* [38], and *Bacillus* spp. [25]. The described genotype-specific association seems to be in agreement with the results of our findings based on discriminant analysis because its model, employing as a grouping variable the cultivar either alone or in combination with the variables BCA and pathogen presence, showed a quite good differentiating ability, expressed by low $\lambda_w$ between 0.06205 and 0.6078 in dependence on the used dataset and particular grouping (Table 3). The resolution was better when the greenhouse part of the data was used, which included three strawberry cultivars, and increased with the addition of the other factors (presence of BCA and of the pathogen) into the discriminant model. Similar accuracy improvement, with an increased number of included factors, was also visible in the data from the cultivating room, where only two strawberry cultivars were tested; however, the total differentiating ability in comparable cases was lower. The differences in the effects of BCA treatments among particular cultivars were also obvious in the results of the PCA (Table 4, Figure 1, Supplementary Figure S1). In the case of the cultivar Karmen, the comparisons of infected and uninfected plants in all variants of BCA treatment resulted in no differences (Table 4), while in the cases of Sonata and Wendy, only some of the BCAs caused such a low differentiation. The differences were also obvious between the effects of different BCAs whose active component is the same species, such as *C. rosea* isolate 156, Gliorex, and Clonoplus (*Clonostachys rosea*) or Gliorex and Xilon GR (*Trichoderma asperellum*). Since Gliorex contains a mix of *C. rosea* and *T. asperellum*, the cause of the difference in effect in comparison to both Clonoplus and Xilon GR could be attributed to the antagonism of those two components; this was documented as being possible in some environmental conditions [32]. Similarly, differences in effect were found among the isolate of *C. rosea* 156 and Clonoplus, but there are no differences in the composition of the active component among them that would explain the different effects. The influence of the particular genotype of the BCA tested, as well as that of the host plant, thus seems to be important for the establishment of a relationship that would effectively protect the plant against the pathogen.

## 5. Conclusions

It was confirmed that the majority of the tested BCAs have an influence on the growth and health of strawberry plants. The majority of the BCAs except for Integral Pro and the isolate of *Pseudomonas* sp. had a significant positive impact on plant growth in the absence of the pathogen, while in its presence the pathogen always caused worsening of the growth. The extent of deterioration of the plants was significantly dependent on the BCA used; some of the tested BCAs even decreased the health of plants, at least in some characteristics (Integral Pro, Contans, *Clonostachys rosea* isolate 156, *Pseudomonas* sp. isolate), while only Polyversum unambiguously decreased the negative effect of the pathogen on the plant. Polyversum exhibited a strong stimulating effect on plant growth, predominating above the negative influence of the pathogen, which we presumed to be the main

mode of action, although other modes of action cannot be excluded. The result of the interaction in the complex of plant, pathogen, and antagonist seems to be dependent on environmental factors and the particular genetic features of all three interaction participants. The unambiguous identification of the environmental, genetic, and other factors most influencing the effect of the BCAs and the environmental conditions in which the BCAs are useful should be the next prospects of the research.

**Supplementary Materials:** The following are available online at www.mdpi.com/2077-0472/11/11/1086/s1, Figure S1: The results of the principal component analysis comparing the plants infected by *Phytophthora cactorum* (**P**) with uninfected control plants (**C**), where all the plants were treated by tested BCAs. The plots depict the similarities/dissimilarities between these two variants in each BCA treatment; the information of the cultivating space used is given. The clear differentiation of the clouds of **P** and **C** points in charts is interpretable as the result of different growth and health of plants as a consequence of pathogen infection. Therefore, such tested compounds cannot be considered as agents providing plant protection in this particular case. The overlap of these two clouds means low or no differentiation between plants caused either by the protective effect of the BCAs on infected plants or by the negative influence of BCA alone, which decreases the plant growth on a similar scale as a pathogen. The variable differentiation is visible on the charts documenting the different abilities of tested BCAs to protect strawberry plants against *P. cactorum*.

**Author Contributions:** M.P. formulated aims of the paper, performed laboratory works and pathogenic tests, participated in measurements and data recording, performed the statistical evaluation of data, and compiled and finalized the manuscript; A.H., J.W. and M.M. participated in the choice of BCAs and their optimization for experiments, performed laboratory works and pathogenic tests, and participated in measurements and data recording; M.Z. participated in the design of experiment and verified results and conclusions. All authors have read and agreed to the published version of the manuscript.

**Funding:** The work was supported by the project of the Ministry of Agriculture of the Czech Republic QK1710377 and by institutional support MZE-RO0418.

**Institutional Review Board Statement:** Not applicable.

**Informed Consent Statement:** Not applicable.

**Data Availability Statement:** The data recorded in this study are available on request from the corresponding author.

**Conflicts of Interest:** The authors declare no conflict of interest.

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
