# Peer review of "A Comparison of the Ability of Some Commercially Produced Biological Control Agents to Protect Strawberry Plants against the Plant Pathogen Phytophthora cactorum"

_agriculture, doi:10.3390/agriculture11111086_

Round 1

Reviewer 1 Report

This paper describes two experiments (one under controlled conditions, and the other in greenhouse) but with a nearly identical experiment design, testing the effect of several biocontrol organisms on strawberry crown rot. However, the paper is far too long for the actual results presented; particularly the discussion section.
I have particular concerns about the experiment design (or the lack of clear description of the design) and data analysis or presentation.
The authors only said that there were 10 pots per treatment, and there were 24 or 21 treatments (3 cultivars, 7 BCAs, and control). However, it is not clear (1) whether all the treatments were conducted at the same time, (2) what is the experiment design, e.g., was the individual pot of all the treatments randomised? These two questions have significant impact on how the data were to be analysed.
The manuscript is about the effect of BCA on crown rot. However, the authors did not even attempt to assessment the actual disease development or assess whether there is crown tissue discolouring [indicating possible pathogen infection]. What is the reason for this?
Given the data set obtained, it should be straightforward to analyse the data with multivariate analysis of variance, testing the effects of biocontrol organisms and assessing whether biocontrol effect varies with cultivars. This is a simple extension of ANOVA to test the main effects of BCAs and cultivars, and their interactions. 
However, the authors used the discriminant function analysis for various pairwise comparisons (i.e. two groups). By doing so, it lost much of statistical powers as all the data are not analysed together (of course, assuming that the experimental design allows the data to be analysed together). Furthermore, such an analysis was not able to test whether biocontrol efficacy varies cultivars - this is an important question and has impact on how biocontrol products are used in practice. 
Next the authors used PCA for the present data set. PCA is an exploratory data analytical method and is not able to provide formal hypothesis testing (.e.g. whether BCAs have an effect). I failed to see any need for this.

Author Response

Dear reviewer,

On behalf of whole authors collective I would like to express our thanks for your valuable recommendations, all of which we thoroughly considered and used to improve the quality of whole manuscript. Our responses to each of yor suggestions and questions I placed directly next to them in the text bellow.

The best regards

Matěj Pánek, corresponding author

R1: This paper describes two experiments (one under controlled conditions, and the other in greenhouse) but with a nearly identical experiment design, testing the effect of several biocontrol organisms on strawberry crown rot. However, the paper is far too long for the actual results presented; particularly the discussion section.
I have particular concerns about the experiment design (or the lack of clear description of the design) and data analysis or presentation.
The authors only said that there were 10 pots per treatment, and there were 24 or 21 treatments (3 cultivars, 7 BCAs, and control). However, it is not clear (1) whether all the treatments were conducted at the same time, (2) what is the experiment design, e.g., was the individual pot of all the treatments randomised? These two questions have significant impact on how the data were to be analysed.

Response: The Discussion part of the manuscript was shortened, some parts of the discussion were used in Introduction, of which deepening was reccomended by other reviewer.

We considered both parts of experiment (greenhouse and cultivating rooms) as a partial tests of one experiment, which is also visible in statistical analyses used. This design made it possible to consider also the influence of the environment on the efficiency of BCAs on infection. Therefore all experiments were performed in exactly the same time and whole settings except of cultivating space. In response to your suggestions we tried to better explain the parameters of experimental design and particularised it directly in the manuscript.

R1: The manuscript is about the effect of BCA on crown rot. However, the authors did not even attempt to assessment the actual disease development or assess whether there is crown tissue discolouring [indicating possible pathogen infection]. What is the reason for this?

Response: We dissected all crowns, cut them across and made the photograph picture of each of them, i.e. we have hundreds of such pictures. In addition, we also tried to reversely isolate the pathogen from this section, but the success was quite variable, independently on the presence of the necrotic lesion, i.e. we succesfully reisolated the pathogen from some of rhizomes without the visible necrosis, while many of reisolations from the obviously necrotic tissues were unsuccessfull. The success of reisolation depends also on other factors than only on presence of pathogen in tissues. To our knowledge, the lession  or discolouration may be present, but this is not the rule, and its presence strongly depend on the time the pathogen reached this part of tissues, it is even suspected that young infected crowns appear symptomless for up to several months before discolouration in the crown first appears in field condition. Therefore we decided to disregard this part of experiment as not very reliable and used undirect but more exactly findable variables.

R1: Given the data set obtained, it should be straightforward to analyse the data with multivariate analysis of variance, testing the effects of biocontrol organisms and assessing whether biocontrol effect varies with cultivars. This is a simple extension of ANOVA to test the main effects of BCAs and cultivars, and their interactions. 
However, the authors used the discriminant function analysis for various pairwise comparisons (i.e. two groups). By doing so, it lost much of statistical powers as all the data are not analysed together (of course, assuming that the experimental design allows the data to be analysed together). Furthermore, such an analysis was not able to test whether biocontrol efficacy varies cultivars - this is an important question and has impact on how biocontrol products are used in practice. 
Next the authors used PCA for the present data set. PCA is an exploratory data analytical method and is not able to provide formal hypothesis testing (.e.g. whether BCAs have an effect). I failed to see any need for this.

Response: In agreement with your suggestions, we initially also pondered to use MANOVA as a useful multicriterial method for data evaluation, but in paralell we also tried to use the Discriminant analysis. Both methods used all variables and resulted by the same basic finding, that there are significant differences between variants in majority of the variables settings used. The same definition of  variants alone, which is the crucial point for both analysing methods, was used in both types of analysis. As we described in the Materials and Methods part of manuscript, not only one settings of tested groups was used, what is detailed in Tab. 3 (new numbering). In total, analysis of thirty one meaningful combinations of used partial datasets (based on cultivating space and tested cultivars; 10 partial datasets)  and used grouping (based on independent variables – cultivar, pathogen, BCAs; 8 groupings) were performed by Discriminant analysis. The most important result of these analyses was the value of λW expressing the differentiation ability of the tested model and identification of variables significantly important for the discrimination. The subsequent comparison was based only on these chosen variables. In MANOVA, this last step correspond to post-hoc tests, which are the most critical part of our case of MANOVA. If the combination of independent variables “pathogen” / “BCAs” / “cultivar” are used, the results of post-hoc tests are the matrices as large as 72×72 cells for each of ten dependent (measured) variables and for  each of the independent variable combinations, where some (and not the same in all matrices) of the differences between variants are significant. To create some definite conclusion based on such results is not possible. The presence of pathogen as well as of BCAs also influences differently various parts of plants which make the comparison just more intricate. In addition, these comparisons compares only the infected plants with these uninfected in the frame of each BCA variant. But, as we also made effort to explain in manuscript, the effect of BCAs alone on plants without pathogen is important; however, it is hard to distinguish it from the effect of pathogen when they are inoculated together on one plant; MANOVA does not offer the tool for distinguishing them. We assume, that regarding these mentioned reasons the MANOVA generally and post-hoc tests in particular are not the most convenient tools for our purposes and instead, we decided to use Discriminant analysis to identify the most important variables for distinguishing and use only their averages for subsequent comparison. To our knowledge, the discriminant analysis use the same techniques and principles as MANOVA, therefore the overall significance of the results should be comparable.

Regarding the PCA analysis results, we used them mainly as a form of the results visualisation, which are well intelligible comparing the complex results of discriminant analysis. Although we evaluated the differentiation in charts, the conclusions of all tests are based mainly on statistically significant results of Discriminant analysis. However, the results of thess PCA brings the useful information of the part of variability of the data associated with each used BCA (including variants with and without pathogen). In the discussion we also accented, that this analysis alone does not offer the proof of unambiguous ability of BCAs to protect plants (Lines 381 – 385).

Reviewer 2 Report

Here are some minor revisions to look at:

  1. The introduction needs more background.
  2. Page 2, Line 86: Where is the pathogen coming from? Did you get it from a company, from the field? What about the pathogen characterization? Did you maintain the pathogen in a fungal collection? Did you register with a code at the NCBI? IDs included in Table 1 are not the ones from NCBI. Sequencing the ITS will be enough to assure the strain you are working with.
  3. Page 2, Table 1: verify decimal separation, use dots instead of commas.
  4. Page 5, Table 2: verify decimal separation, use dots instead of commas. Correct headers.
  5. Page 7, Table 3. Correct headers.
  6. Table 2 must be moved to the “Discriminant analysis” subsection of Results. It is hard to follow the results going back and forth to the Material and Methods section to look at the table. In addition, Table 2 shows the results themselves so it has to be there.
  7. Table 4 should be placed before Figure 1 to follow the order of the results
  8. Page 8, Line 247: information in this line should be presented more clearly.
  9. Page 9, Line 272: the figure legend contains information that must be presented in the result section.
  10. Page 10, Table 4: verify decimal separation, use dots instead of commas.
  11. Page 11, Line 298: Define R:S
  12. Supplementary figures S1 need a figure legend.

  13.  

     Figure legend for Table 4 contains unnecessary information to understand the table itself. The extra information could be placed in the result section.

  14.  

     Figure 1 and Supplementary figures S1 are presented in back and white. I think to better understand the information from those figures others colors are needed.

  15.  

     Results from table 4 are presented in the text before table 1, so the order of them in appearance should be changed.

  16.  

     The principal component analysis section of Results needs to include a detailed discussion of the results shown in Figure 1.

  17.  

     The data presented in Table 5 as a percentage is discussed along with the result section 3.3. However, from my point of view more explanatory information of the meaning of the percentages is needed.

  18. In the attached pdf version of the manuscript, you could find more comments to take into consideration

Author Response

Dear reviewer,

On behalf of whole authors collective I would like to express our thanks for your valuable recommendations, all of which we thoroughly considered and used to improve the quality of whole manuscript. Our responses to each of yor suggestions and questions I placed directly next to them in the text bellow.

The best regards

Matěj Pánek, corresponding author

  1. The introduction needs more background.

Response: We decided to move some parts of Discussion, which enlarged this part, to the Introduction. The shortening of Discussion was also the demand of other reviewer. We hope this movement improved the structure of whole text.

  1. Page 2, Line 86: Where is the pathogen coming from? Did you get it from a company, from the field? What about the pathogen characterization? Did you maintain the pathogen in a fungal collection? Did you register with a code at the NCBI? IDs included in Table 1 are not the ones from NCBI. Sequencing the ITS will be enough to assure the strain you are working with.

Response: We completed the corresponding information into tables and the text. Since all isolates of pathogen were identified and used in another published works, we originally neglected to give the information in the actual text. This mistake has been thus removed now. Thank you for this suggestion.

  1. Page 2, Table 1: verify decimal separation, use dots instead of commas. OK
  2. Page 5, Table 2: verify decimal separation, use dots instead of commas. Correct headers. OK
  3. Page 7, Table 3. Correct headers. OK
  4. Table 2 must be moved to the “Discriminant analysis” subsection of Results. It is hard to follow the results going back and forth to the Material and Methods section to look at the table. In addition, Table 2 shows the results themselves so it has to be there.

Response: The table was moved in accordance to your demands. However, such movement forced to remove the mention of Table 2 (newly - Table 3) from the MaM section, since its occurence there would lead to the original numbering (the first mention of the table in text). This change slightly decreased the intelligibility of explanation of the set-up of the data statistical evaluation, which is also indicated in this table. Since this problem has not ideal solution, we accepted your suggested one.

  1. Table 4 should be placed before Figure 1 to follow the order of the results. OK
  2. Page 8, Line 247: information in this line should be presented more clearly.

Response: We tried to explain the set up of the statistical evaluation in better way. Hope, this time we have been more successfull.

  1. Page 9, Line 272: the figure legend contains information that must be presented in the result section.

Response: The mentioned part was moved into the Results section.

  1. Page 10, Table 4: verify decimal separation, use dots instead of commas. OK
  2. Page 11, Line 298: Define R:S

Response: We newly add the definition - the mention on page 5, line 196.

  1. Supplementary figures S1 need a figure legend.

Response: The legend was completed and intelligibility of all figres was improved by use of two colours and shapes of the points.

  1.   Figure legend for Table 4 contains unnecessary information to understand the table itself. The extra information could be placed in the result section.

Response: The demanded information was placed into text, into first paragraph of part 3.2 Principal component analysis

  1. Figure 1 and Supplementary figures S1 are presented in back and white. I think to better understand the information from those figures others colors are needed.

Response: see the point 12. response.

  1.   Results from table 4 are presented in the text before table 1, so the order of them in appearance should be changed.

Response: We were not successful to find out the mention of the results of table 4 before table 1. Tab. 1 is part of the MaM section, thus the presentation of results there seems does not have sense.

  1.   The principal component analysis section of Results needs to include a detailed discussion of the results shown in Figure 1.

Response: The discussion of the results of this part was moved from the Figure 1 description into Results PCA part. The caption of the figure has been appropriately shortened.

  1.   The data presented in Table 5 as a percentage is discussed along with the result section 3.3. However, from my point of view more explanatory information of the meaning of the percentages is needed.

Response: The paragraph briefly detaling this issue was add into the introduction of the section 3.3.

  1. In the attached pdf version of the manuscript, you could find more comments to take into consideration

The part Conclusions was completed by few the most important findings of our research and by the possible outline as follows from these findings.